# FROM CONTEXT TO CONCEPT: CONCEPT ENCODING IN IN-CONTEXT LEARNING

## ABSTRACT

Humans distill complex experiences into fundamental abstractions, enabling rapid learning and adaptation. Similarly, autoregressive transformers exhibit adaptive learning through in-context learning (ICL), which begs the question of how. In this paper, we propose **concept encoding-decoding mechanism** to explain ICL by studying how transformers form internal abstractions in their representations. On synthetic ICL tasks, we analyze the training dynamics of a small transformer and report the coupled emergence of concept encoding and decoding. As the model learns to encode different latent concepts (e.g., "Finding the first noun in a sentence.") into distinct, separable representations, it conditionally builds decoding algorithms and improve its ICL performance. We validate the existence of this mechanism across pretrained models of varying sizes (Gemma-2 2B/9B/27B, Llama-3.1 8B/70B). Further, through mechanistic interventions and controlled finetuning, we demonstrate that the quality of concept encoding is causally related and predictive of ICL performance. Our empirical insights shed light into better understanding the success and failure modes of large language models via their representations.

## 1 INTRODUCTION

Throughout history, humans have made sense of the world by distilling complex experiences into fundamental abstractions, such as physics and mathematics. These mental models enable us to learn quickly, predict outcomes, and adapt to new situations. In artificial intelligence, autoregressive transformers are beginning to exhibit similar capabilities. Through in-context learning (ICL), they adapt to new tasks without parameter updates, suggesting they might also be forming internal abstractions (Raventós et al., 2024; Hong et al., 2024; Zheng et al., 2024b; Kumar et al., 2024).

Hendel et al. (2023); Merullo et al. (2023); Todd et al. (2023) introduce a mechanistic perspective on how pretrained LLMs represent the latent concepts underlying the ICL task as vectors in their representations. They empirically demonstrate that these task-specific vectors can trigger the desired ICL behavior in many cases, with the effectiveness varying across tasks. Although an impactful first step, there still remains unanswered questions of why these task vectors exist in the first place and why the effectiveness varies by task. This necessitates a deeper mechanistic understanding of the internal abstraction behavior of LLMs, which could encompass the findings of task-specific vectors and various aspects of ICL.

In our work, we propose the **concept encoding-decoding mechanism** as the origin of internal abstraction behavior. To study the emergence of abstractions during pretraining, we train a small transformer on a mixture of sparse linear regression tasks. We find that **concept encoding** emerges as the model learns to map different latent concepts into *distinct, separable representation spaces*. This geometric structuring of the representation space is coupled with the development of concept-specific ICL algorithms – namely, **concept decoding**. Importantly, we see that the emergence of the two-stage process coincides with one another, implying mutual dependence between the two. Through causal analysis, we demonstrate that the model associates different algorithms to different learned concepts and that ICL happens through the two-step process.

We demonstrate the validity of the concept encoding-decoding mechanism across different pretrained model families and scales (Llama-3.1-8B/70B and Gemma-2 2B/9B/27B) on more natural

ICL tasks, such as part-of-speech tagging and bitwise arithmetic. We show that large language models (LLMs) trained on diverse data also exhibit concept encoding behavior. With more in-context examples, LLMs map the inputs to increasingly separable representation spaces, clustered by the latent concepts. Moreover, leveraging insights from the synthetic experiments, we demonstrate that the decodability of the concepts from representations is predictive of downstream ICL performance. We establish a causal relationship between the quality of encoding and ICL performance through mechanistic intervention and controlled finetuning experiments.

Our main contributions are as follows:

1. We first study the emergence of task vectors by training a small transformer on a synthetic ICL task (§3.3) and propose **concept encoding-decoding mechanism** to explain the emergent behavior for learning to solve ICL tasks. We observe that earlier layers of the model learn to encode the latent concept, whereas the latter layers conditions the algorithm on the inferred concept. Interestingly, the emergence of the two-stage process is coupled, implying a mutual dependence.

2. We introduce Concept Decodability (CD), a quantitative metric that predicts downstream ICL performance in pretrained LLMs (§4.2). We demonstrate our framework's generality across tasks, model families, and scales (Llama 3.1 8B/70B, Gemma 2B/9B/27B)

3. We establish the causal relationship between CD and ICL performance in pretrained LLMs through mechanistic intervention (§4.1) and controlled finetuning (§4.3).

4. We offer a new perspective on mechanistically understanding how the model internalizes the learning signal of more in-context examples, finetuning, and prompting (§5.1) through the lens of concept encoding-decoding.

## 2 RELATED WORK

**Mechanisms of ICL.** Astounded by LLMs' ability to perform ICL, many have proposed theories to understand the mechanisms of ICL. Some (Dai et al., 2023; von Oswald et al., 2023; Ahn et al., 2024; Akyürek et al., 2024) have proposed that LLMs, with linear attention (Katharopoulos et al., 2020), can implement stochastic gradient descent to perform ICL. Other works (Xie et al., 2021; Wang et al., 2024; Ye et al., 2024) have presented a Bayesian framework to theoretically explain the workings of ICL. This view implies that the model implements a two-stage algorithm to estimate the posterior $P(z|\mathcal{D})$ and the likelihood $P(y_*|x_*, \mathcal{D})$. In this work, we adopt this framework and demonstrate how the model implements it through its intermediate representations. More specifically, we study the emergence of concept encoding – building separable representations for different latent concepts.

**Task Vectors.** Todd et al. (2023) and Hendel et al. (2023) identify task/function vectors that can induce desired ICL task behavior (e.g., object-color mapping) even at zero-shot. Although motivated by their work, we propose *concept encoding* in place of the term 'task vector' because of the limited scope under which it is valid. Previous studies (Pan et al., 2023; Wei et al., 2023b; Min et al., 2022) have found ICL tasks that are word-to-word mapping (e.g., object-color, English-French) are not in fact task learning but task retrieval that uses semantic priors of the model. Moreover, Zheng et al. (2024a) demonstrated that the functions' representation vectors are distributed over multiple tokens for more complex tasks. Therefore, we use 'concept encoding' to refer to a broader phenomenon of building separable representations for distinct latent concepts.

**Latent Concepts in Language Model Representations.** Several studies (Dalvi et al., 2022; Merullo et al., 2023) have examined how language models encode concepts in their representations. In autoregressive LLMs, notions like truthfulness (Marks & Tegmark, 2023) and time and space(Gurnee & Tegmark, 2024) have been shown to be linearly separable representations. Sparse autoencoders (Bricken et al., 2023; Cunningham et al., 2023) have revealed highly interpretable features that emerge and grow with scale in LLMs. Beyond the identification of these concepts, our work aims to answer how such concepts emerge in the representation of LLMs and how they causally relate to ICL performance.

**Mechanistic Interpretability.** To study the causal relationship between the accuracy of concept encoding and downstream ICL performance, we adopt causal mediation analysis techniques from Geiger et al. (2020); Vig et al. (2020); Todd et al. (2023); Heimersheim & Nanda (2024); Merullo et al. (2024). We specifically use the method of activation patching, where we replace the activations of an immediate layer from a sample with another. This technique allows us to demonstrate that transformers implement different algorithms conditioned on the inferred concepts.

# 3 UNDERSTANDING IN-CONTEXT LEARNING

## 3.1 NOTATION AND BACKGROUND

We focus on ICL problems, where the goal is to predict $y_*$ from a query $x_*$, given some in-context examples $\mathcal{D} = \{(x_i, y_i)\}_{i=1}^n$. Each problem shares a latent concept $z$ that links inputs $x$ to outputs $y$. For instance, in an ICL task where latent concept is object-color mapping, we provide demonstrations like (apple, red), (banana, yellow), and (grape, purple), and then ask for what comes after (lemon, ?). We employ this parameterization to accommodate latent concepts varying in complexity, from simple function regression problems (Garg et al., 2022; von Oswald et al., 2023; Li et al., 2023) to POS tagging (Blevins et al., 2022; Banko & Moore, 2004) and arithmetic (He et al., 2024).

## 3.2 THEORETICAL FRAMEWORK

Of the many different frameworks (Bai et al., 2024; Min et al., 2022; von Oswald et al., 2023; Akyürek et al., 2024) to understand the workings of ICL, we adopt the Bayesian view (Xie et al., 2021; Mittal et al., 2024; Wang et al., 2024; Ye et al., 2024). It proposes that transformers implicitly infer the latent variable $z$ underlying the demonstrations and apply it to generate an answer. More formally,

$$p(y_* \mid x_*, \mathcal{D}) = \int_{\mathcal{Z}} P_\theta(y_*|x_*, z) P_\theta(z|\mathcal{D}) \, dz$$

This framework suggests ICL is a two stage process. First, *latent concept inference*. Latent concept $z$ is approximated from $\mathcal{D}$ through the distribution $\hat{z} \sim P_\theta(z|\mathcal{D})$. Second, *selective algorithm application*. The model applies an algorithm conditioned on $\hat{z}$ to predict $y_*$ as given by $P_\theta(y_*|x_*, \hat{z})$.

Although theoretically compelling, it was not until recently that Hendel et al. (2023); Todd et al. (2023); Merullo et al. (2023) showed empirical evidence of models encoding the latent concepts in the intermediate representations. They illustrate that these representative vectors are then decoded and trigger the desired ICL task behavior. With this simple analogy to an encoder-decoder, we begin our investigation into the following questions:

1. How does the concept encoding and decoding behavior emerge in the model over training and how do they interplay?
2. How is the model's ability to accurately infer the latent concepts related to downstream ICL performance?

## 3.3 MOTIVATION: SYNTHETIC EXPERIMENTS

We train a small transformer on a synthetic ICL task and demonstrate that concept encoding and decoding emerges simultaneously during training. Through causal analysis, we show that, as the models "discovers" a latent concept by building a distinct representation from the others, it associates the concept with different decoding algorithms. Finally, we propose the concept encoding-decoding mechanism that encompasses these findings and serve as the core theory throughout the remainder of our study.

**Task.** We compose our task as a mixture of sparse linear regression tasks. We follow the conventional linear regression setup from Garg et al. (2022); von Oswald et al. (2023) and construct the input-output pair $(x_i, y_i)$ by sampling $x_i \sim \mathcal{N}(0, \boldsymbol{I_D})$ and $y_i = W^T x_i + \epsilon_i$, where $W$ is randomly generated from a standard normal distribution, $\mathcal{N}(0, \boldsymbol{I_D})$, and $\epsilon_i \sim \mathcal{N}(0, \sigma^2)$. We, however, add

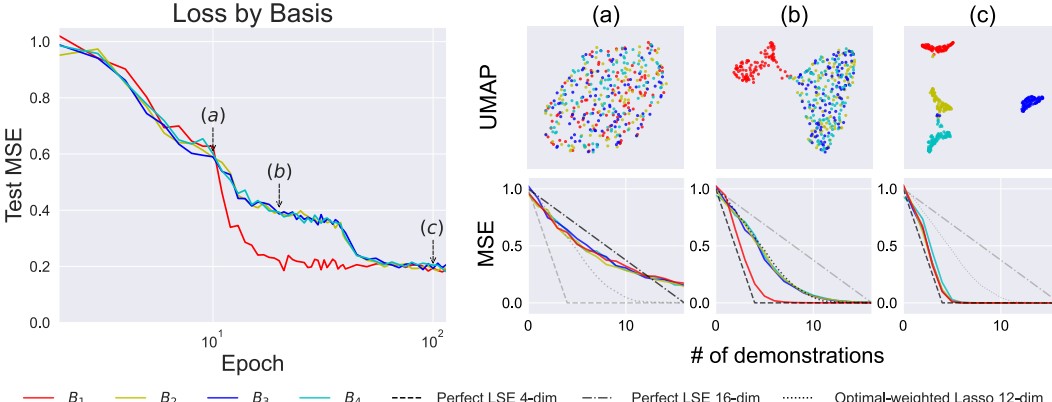

Figure 1: Coupled emergence of concept encoding and conditional decoding algorithms in mixture of sparse linear regression. The loss curve on the left-hand side shows different convergence dynamics per basis and show three phases of descent, which we mark with (a), (b), and (c). On the right-hand side, we plot the geometric changes in the representations and how they separate by basis at these marked points. These points coincide with the algorithmic switching behavior.

sparsity constraints to $W$ with the sparsity pattern represented by the basis $B_k$. Each $B_k$ has a rank of $r$. In other words, the basis chooses the dimensions of $W$ to turn on and off. The basis is sampled uniformly from $\mathcal{B} = \{B_1, B_2, B_3, B_4\}$ and each basis is non-overlapping and orthogonal to each other. By default, we set $D = 16$ and $r = 4$. By adding this layer of latent concept of $\boldsymbol{B}$, we can explicitly control and interpret the latent concepts, and analyze their representations.

**Model and Training.**   We train a 12-layer GPT-2 architecture transformer (Radford et al., 2019) with an embedding dimension of 128 and 8 attention heads. We train the model to minimize mean squared error (MSE) loss over the sequence of context length 20. We run 5 different random seeds for training and report observations that generalize across the runs. We detail the experimental setup further in Appendix C.2.

**Theoretical Error Bounds.**   The error bounds of our task depend on whether the model learns to infer the underlying bases. If the model learns to infer the bases, then the model can achieve $r$-dimensional regression, where the MSE approaches 0 with $r$ in-context examples. If not, the model, in the worst case, can perform $D$-dimensional regression with $r$-sparsity, which has a longer tailed error curve that approaches 0 between $r$ and $D$ in-context examples. With these insights, we can better analyze which latent basis the model has learned and the associated algorithm. Note that we define "algorithm" as a class of statistical methods for linear regression, as detailed in Appendix C.1.

**Observation 1: Different Loss Dynamics Per Basis.**  We interestingly observe that each basis, despite having identical task complexities, exhibits different loss descents during training. Figure 1 shows the test MSE averaged over the sequence over training. $B_1$ displays a distinct loss descent dynamic, undergoing an abrupt drop at epoch 10. In contrast, the other three bases, $B_2$, $B_3$, and $B_4$, exhibit correlated loss descent dynamics, with two smaller descents at 10 and 40 epochs. This suggests that the model learns to infer $B_1$ differently and applies selective algorithms.

**Observation 2: Emergence of Separable Representations and Coupled Algorithmic Phase Transitions.** We also analyze the geometry of the intermediate representations at layer 5 to question how the model may be encoding the latent bases. Surprisingly, at the three points of descent (a, b, c) marked in Figure 1, the model gradually builds separate representations for the different bases as shown in the UMAP visualizations. At point (a), the three bases are clustered together and the model's algorithm resembles a 16-dimensional weighted LASSO regression. As $B_1$ separates out at point (b), the model starts to leverage the inferred basis to switch to a 4-dimensional regression. At point (c), when all four classes are separable, the model converges to the optimal 4-dimensional regression. This observation suggests that model encode latent concepts into separated representations to conditionally apply decoding algorithms.

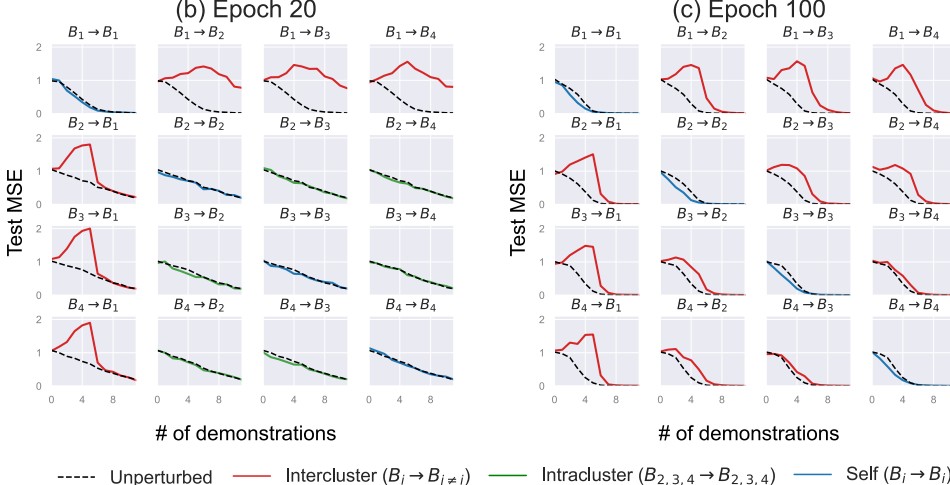

Figure 2: Causal analysis by perturbation. On the left are perturbation results at epoch 20, when the latent concepts' representations are semi-separate ($B_1$ and $B_{2,3,4}$). Intracluster refers to $B_{2,3,4}$. At this stage of training when there are only two clusters of representations, there only exists two decoding algorithms as well. On the right are results at convergence, when the latent concepts' representations are fully separable. In this case, each $B_i$ follows a different algorithm and patching the activations of any other basis than itself increases the loss noticeably. On the other hand, self-perturbation improves ICL performance.

**Causal Relation between Concept Encoding and Performance.** We conduct perturbation analysis to validate that the model conditionally applies decoding algorithms based on the separated representations. Given an input of a source basis, we patch the activation of layer 5—representing the residual stream of the transformer layer—with the mean activation of a target basis and analyze whether it will improve or degrade performance. When the source is equal to the target (*self-perturbation*), the patching should help the model identify the basis and improve performance. Otherwise, it should hinder correct basis inference and therefore performance. We perform this analysis at points (b) and (c) from Figure 1, when the latent concept representations are semi and fully separable.

In Figure 2, we present the perturbation analysis at point (b) on the left. In this case, $B_{2,3,4}$ forms one cluster and $B_1$ another. We observe that all the self-perturbations along the diagonal and intracluster ($B_{2,3,4}$) slightly decrease the loss or show no effect. However, when we apply perturbations across different clusters, the loss spikes, indicating that we trigger different decoding algorithms unsuitable for the input sequence. This analysis shows that, because the model was only able to encode two different latent concepts in the intermediate representations, it only learns two classes of algorithms, one for $B_1$ and another for $B_{2,3,4}$.

On the right of Figure 2, we conduct the same perturbation study at convergence, when the model learns to encode all of the latent concepts as distinct representations. Surprisingly, we observe that the model undergoes an algorithmic phase transition of implementing concept-conditioned algorithms. Not only does all the self-perturbation along the diagonal improve performance more noticeably, but also any perturbation to a different basis results in significantly higher losses.

These results altogether draw the picture that a transformer, when trained to perform ICL, gradually learns to encode the latent concepts to separable representation spaces and learns to conditionally apply decoding algorithms *simultaneously*. These observations suggest that concept encoding and decoding are mutually dependent, but whether they reinforce each other needs to be further studied.

### 3.4 CONCEPT ENCODING AND DECODING

We introduce the terms **concept encoding** and **decoding** in this work to capture the emergent phenomenon of a transformer learning to implement a two-stage process to perform ICL, as observed

in Section 3.3. The model learns to encode different concepts into distinct, separable internal representations. Simultaneously, this separation allows the model to develop concept-specific decoding algorithms. We show that there is a mutual dependency between the two mechanisms and that both are required for effectively solving ICL. What this mutual dependency implies, as we explore through the causal perturbation studies, is that the accuracy to which the model encodes and distinguishes the latent concepts – namely, concept decodability – is predictive of downstream ICL performance. We now validate our theory in real-world, pretrained LLMs.

# 4 TOWARDS NATURAL EXPERIMENTS

In this section, we empirically validate the proposed concept encoding-decoding mechanism in pretrained LLMs. Specifically, we test several hypotheses derived from the proposed mechanism, such as whether pretrained LLMs exhibit concept encoding behavior and whether the accuracy of concept encoding can predict ICL performance on more natural tasks.

**Tasks.** We construct two classes of algorithmic tasks – natural language processing and arithmetic – comprising a total of 12 tasks. Within each class, the tasks are designed to be semantically similar, ensuring that the input distributions are alike across tasks. While the underlying latent concepts differ (e.g., different arithmetic operations or linguistic patterns), the surface features of the inputs remain consistent. By keeping the input distributions similar, we can effectively assess the model's ability to infer and encode latent concepts based solely on subtle differences in the data, rather than the input variations. Refer to Appendix E for more details.

*Part-of-Speech (POS) tagging.* We construct a POS tagging (Blevins et al., 2022; Banko & Moore, 2004) dataset from Marcus et al. (1994), consisting of POS tags, such as Noun, Adjective, Verb, Adverb, Preposition, Pronoun. Given an input text and hidden POS tag $z_i$ (e.g., Noun), one needs to output the first word that is of the specified POS tag.

*Bitwise arithmetic.* We construct a bitwise arithmetic dataset consisting of 6 different operators, AND, NAND, OR, NOR, XOR, and XNOR. Given a pair of 5-digit binary numbers and the hidden operator $z_i$ (e.g., AND), one needs to output the resulting binary number after the operation.

For both of these tasks, we create an additional Null class, for which there is no latent concept. In bitwise arithmetic, the Null operator outputs random binary digits, and in POS tagging, the Null class pairs the input sentences with a randomly selected word. This task helps us identify the cases in which the model is confused about the concept.

**Model.** We verify the existence of our proposed concept encoding-decoding mechanism in models of different families and scales (Gemma-2 2B/9B/27B and Llama-3.1-8B/70B) in Section 4.1 and Appendix E. We continue further analysis with the pretrained Llama-3.1-8B model (Meta, 2024). We do not train this model, except when we study the causal effect of concept decodability by finetuning in Section 4.3. We further detail the experimental setup in Appendix E.

**Evaluation.** We evaluate the performance of the model on different tasks by computing the exact-match accuracy between the generated output under greedy decoding and the ground truth. All of the evaluations assume 4-shots of examples, unless specified otherwise.

**Concept Decodability (CD).** To quantify how well latent concepts can be inferred from representations, we employ a simple $k$-Nearest Neighbor (k-NN) classification metric. Inspired by prior studies using linear probes (Rimanic et al., 2020; Alain & Bengio, 2018), we assess whether the latent concepts can be extracted in a simple manner from their representations. Specifically, we use the representations of the token immediately before $y_*$ at a chosen layer and predict the latent concept by majority voting among its $k$ nearest neighbors ($k = 10$, $N = 100$).

## 4.1 CONCEPT ENCODING-DECODING IN PRETRAINED LLMS

**Hypothesis**: Concept encoding-decoding behavior exists in pretrained LLMs.

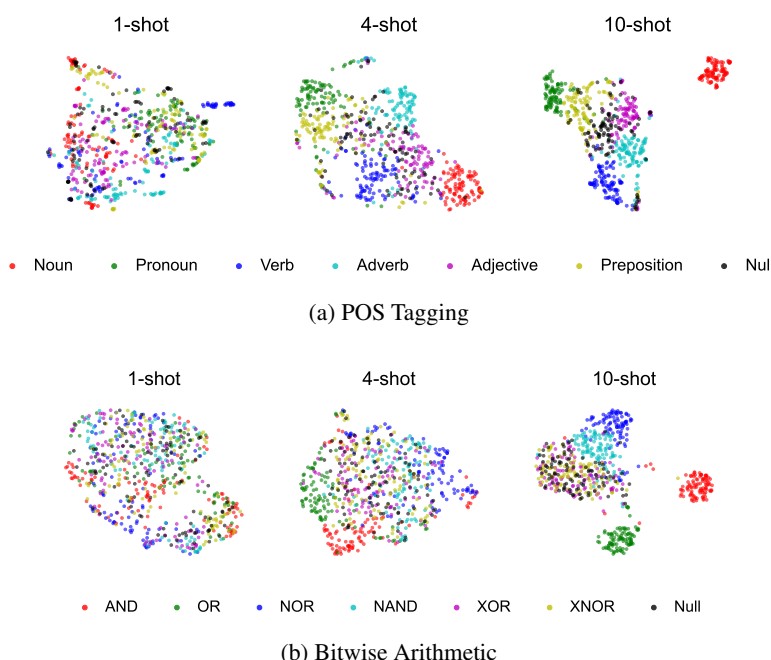

Figure 3: Concept encoding in Llama-3.1-8B. UMAP of the intermediate representations at layers 15 and 13 respectively for POS tagging and bitwise arithmetic with varying number of in-context examples (1, 4, 10-shot). With more in-context examples, the model builds increasingly separable representations clustered by their latent concepts.

We investigate the above hypothesis in two steps: (1) We first examine whether concept encoding occurs in pretrained LLMs; (2) We conduct mechanistic intervention studies to verify that different concept encoding triggers separate decoding algorithms, completing the full study of the concept encoding-decoding mechanism.

**Step 1: Concept Encoding.** We first study whether the concept encoding occurs in pretrained LLMs. We vary the number of in-context examples for the different tasks and visualize the intermediate representations at the middle layers with UMAP in Figure 3. Given only 1-shot, where the model is expected to be confused about the latent concept, all of the representations are clustered and overlap with the Null class, which has no task latent. As examples increase, clustering by latent concepts emerges, becoming clearer by 10-shots. Interestingly, the separation of concepts, such as AND, OR, Noun, and Pronoun, is more pronounced. We conjecture the model likely sees and learns these concepts better during pretraining. However, there remains a few classes like XNOR and XOR in bitwise arithmetic and Adjective and Preposition in POS tagging that overlap with Null. These observations highlight that the model achieves better concept encoding through more demonstrations, and offers an alternative perspective on how more in-context examples mechanistically improve performance. We will further explore this connection between the separability of concepts in the representation space and ICL performance in Section 4.2.

To quantify how the separability of representations translates into the decodability of the latent concepts, we compute the CD scores across the layers. In Figure 4a, we see that the the decodability of the latent concepts peaks in the intermediate layers, suggesting that the models are encoding the latent concepts through separable representations.

**Step 2: Mechanistic intervention study.** Having shown concept encoding in pretrained LLMs, we now conduct mechanistic intervention studies, adapted from Hendel et al. (2023); Todd et al. (2023). We question whether different concept encoding triggers different decoding algorithms and if the two are causally related. If so, helping or hindering the model's ability to infer the latent concept in the given input should improve or degrade its performance in downstream tasks, respectively. We conduct this casual analysis, by patching a layer's output activations with the mean activations

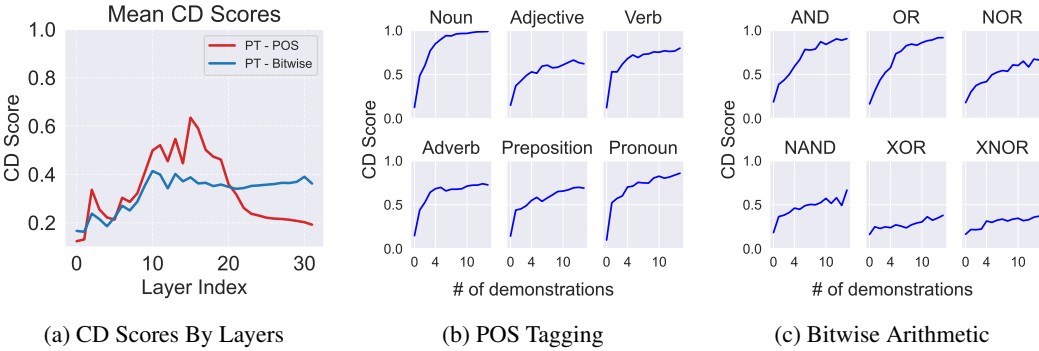

Figure 4: CD Scores by layers and number of demonstrations. (a) Mean CD scores across layers for POS tagging and Bitwise arithmetic with 4-shot in-context examples, showing peak decodability in intermediate layers. (b) For POS tagging and (c) for Bitwise arithmetic, CD scores all increase with the number of demonstrations, but the improvement in CD noticeably varies by task.

of 100 samples with the true latent concept (*positive intervention*) and with the Null latent variable (*negative intervention*). We present the results in Figure 17 of Appendix E.1. For POS tagging (Figure 17a), intervening positively improves performance by $\sim 14\%$ and intervening negatively degrades performance by $\sim 15\%$ on average across the 6 tasks. In bitwise arithmetic, the influence of the interventions is less stark. Positive intervention improves performance by $\sim 2\%$ and negative intervention degrades performance by $\sim 6\%$ on average across all the 6 tasks. Both positive and negative interventions are more effective for tasks whose representation sub-spaces are clearly separated. For tasks whose representation overlap with those of Null, we hypothesize that the model is failing to infer the latent concept, rendering the intervention less effective.

Overall, through these two studies of the geometry of representations and mechanistic intervention, we demonstrate that concept encoding is causally linked to different decoding algorithms and that concept encoding-decoding behavior indeed exists in pretrained LLMs.

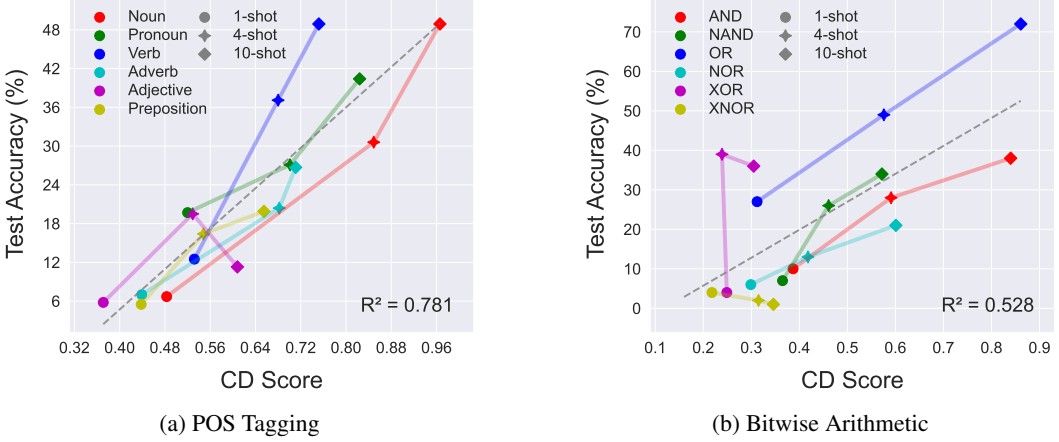

Figure 5: CD score vs ICL performance. We observe a positively correlated trend across most tasks. The grey dashed lines are linear lines of best fit. These results suggest that the accuracy of concept encoding is closely coupled with downstream ICL performance.

## 4.2 PREDICTABILITY OF IN-CONTEXT LEARNING TASK PERFORMANCE

**Hypothesis**: Quality of concept encoding-decoding, measured by CD, is predictive of ICL performance.

We now investigate the second hypothesis of whether the quality of concept encoding-decoding is predictive of downstream ICL performance. If the model is conditionally applying a decoding algorithm to perform the task by first inferring the latent concept, CD and ICL task performance should be closely correlated. To this end, we analyze the relationship between CD and test accuracy by varying the number of in-context examples in Figure 5. In both datasets, we see that, generally, higher CD scores correspond to better performance on the respective tasks. More interestingly, referring back to Figure 3, we remark again that the representations of some classes (Adjective and Preposition in POS tagging and XOR and XNOR in bitwise arithmetic) are mapped to those of the Null class. We notice that this set of classes whose representations overlap with those of Null generally have low task performance and do not improve as much as the others given more demonstrations. We conjecture that the model does not accurately encode latent concepts of those that are overlapped with the Null class representations.

We also test the generality of the predictability of ICL performance from CD across a different model family (Team, 2024) and scales. We conduct the same analysis on Gemma-2 2B, 9B, and 27B and Llama-3.1 70B and present the results in Figure 18 in Appendix E.2. These results demonstrate that the correlation between CD and ICL performance robustly hold across models and tasks. Interestingly, in all of the Gemma-2 family and Llama-3.1 70B models, Noun, Pronoun, and Verb show the clearest signs of concept encoding-decoding behavior, as we saw in the Llama-3.1 8B model. In the bitwise arithmetic task, AND, NAND, OR, and NOR (classes that showed the strongest encoding-decoding behavior in Llama-3.1 8B), also show the strongest signs of concept encoding-decoding behavior across all of these models. Given that many LLMs are trained on similar sources of pre-training data (Soldaini et al., 2024; Gao et al., 2020) (CommonCrawl, Wikipedia, etc), we conjecture that the models may have learned similar encoding-decoding mechanisms for these concepts.

Another natural curiosity that arises is whether this correlation can also be observed during pretraining. Although computationally infeasible to explore this with large-scale pretraining experiments, we demonstrate the correlation between CD and performance by evaluating those across the training iterations of OLMo-7B (Groeneveld et al., 2024) in Figure 8 of Appendix A. Overall, these results demonstrate that the model's ability to infer the correct latents is generally correlated to its ICL task performance.

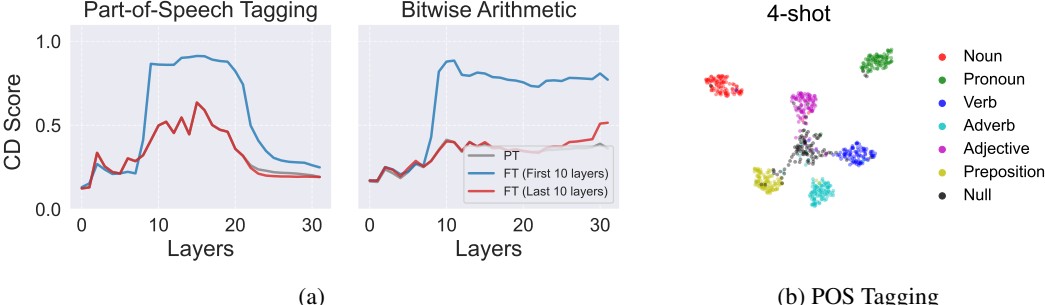

(a)                                                                                            (b) POS Tagging

Figure 6: (a) CD scores across layers for POS and arithmetic after finetuning the first 10 and last 10 layers, at 4 shots. While finetuning (FT) the last 10 layers has minimal effect on the CD scores, finetuning the first 10 layers significantly increases the concept decodability. This phenomenon is accompanied by noticeable improvement in ICL performance. PT denotes the pretrained LLM. (b) UMAP visualization of FT first 10 layers. We illustrate that the increased CD scores correspond to a clear cluster of the representations by latent concepts.

### 4.3 INVESTIGATING THE CAUSAL EFFECT OF CONCEPT ENCODING BY FINETUNING

> **Hypothesis**: In transformers, earlier layers learn to encode concept, whereas the latter layers condition the algorithm on it. Thus, finetuning only the earlier layers can improve concept encoding and thus will be more effective for improving ICL performance than finetuning only the latter layers.

To further investigate the causal importance of concept encoding for downstream ICL performance, we perform two types of finetuning: only the first 10 layers versus only the last 10 layers. We

previously found that concept encoding occurs in the middle layers (layer 15 for POS tagging and layer 13 for bitwise arithmetic). Finetuning only the last 10 layers restricts the model from learning to encode latent concepts in intermediate representations. As illustrated in Figure 6, finetuning the last 10 layers barely changes their CD scores from the pretrained model. In contrast, finetuning the first 10 layers significantly improves the CD scores and aligns the representation subspaces with the inferred latent concepts. This improvement in CD scores directly translates to significantly better ICL task performance. With 4-shot examples, finetuning the first 10 layers outperforms finetuning the last 10 layers by $\sim 37\%$ in the POS task and 24% in bitwise arithmetic. In the bitwise arithmetic task, finetuning the first 10 layers achieves near-perfect accuracy for all tasks except XNOR, whose representations overlap with those of Null.

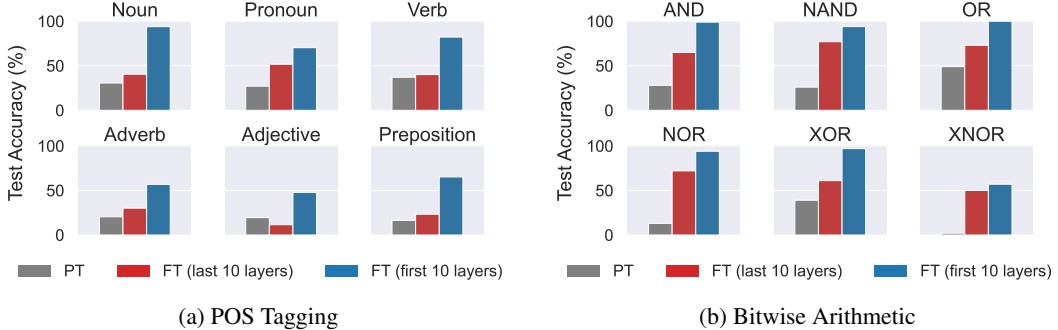

(a) POS Tagging          (b) Bitwise Arithmetic

Figure 7: ICL test accuracy at 4 shots across 12 tasks in POS and arithmetic after finetuning (FT) the first 10 and last 10 layers. When restricting the model's ability to encode latent concepts in its intermediate representation (finetuning last 10 layers), the model fails to fully align its representations for learning the latent concepts and falls behind the performance of finetuning the first 10 layers.

## 5 DISCUSSION

### 5.1 PIECING IT ALL TOGETHER: IN-CONTEXT EXAMPLES, PROMPTING, AND FINETUNING

Our study reveals that enhancing concept encoding is a unifying principle that improves in-context learning (ICL) across different strategies. We observe in Sections 4.2 and 4.3 that increasing in-context examples and finetuning facilitate building separable representations by their latent concepts. Many have also noted that prompting (Wei et al., 2023a) is a simple and effective method of improving in-context performance. Thus, we experiment with prompting as part of our curiosity. We question whether providing the underlying concept (i.e., including true labels of bitwise arithmetic) indeed enhances concept encoding and, as expected, performance. As shown in Figures 20 and 21 of Appendix F, prompting in fact improves the concept encoding and performance simultaneously. However, we interpret these results with caution, since the model may be capturing spurious correlations from the input prompt differences.

### 5.2 WHY DO MODELS SUCCEED AT SOME ICL TASKS, BUT NOT OTHERS?

It is yet puzzling how to categorize the types of ICL tasks LLMs can and cannot perform (Qiu et al., 2023; Dziri et al., 2023). An intuitive explanation is that the model can effectively encode the concepts frequently seen during pretraining (Razeghi et al., 2022; Li et al., 2024). In our experiments, we also observe similar patterns where AND and OR were encoded more accurately. However, we aim to provide an alternative perspective to understand the model's success and failure modes. Through the study of the two-stage process, we show that the bottleneck in the model can exist in both levels of concept inference and subsequent decoding algorithm. Therefore, even if the model already learned the algorithm for a NOR operator, if the model cannot clearly distinguish the latent concept from the inputs, it will fail, and vice versa. Perhaps as our experiments suggest, when the model is failing at concept encoding, a different prescription of finetuning only the earlier layers for better representation learning is more beneficial. Ultimately, by detecting the different causes of the failure modes of models, we hope to build more effective, robust strategies to improve them and unravel the mysteries of large models via mechanistic understanding.

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

## A  INVESTIGATING PREDICTABILITY OF ICL TASK PERFORMANCE IN LARGE-SCALE PRETRAINING

Since it is computationally infeasible to conduct large-scale pretraining studies, we leverage the different training checkpoints for OLMo-7B (Groeneveld et al., 2024) to investigate the relationship between concept decodability and ICL task performance on POS tagging. Interestingly, as shown in Figure 8, we observe a correlated emergence of the two variables. This analysis shows that the coupled emergence of concept encoding and decoding algorithms may also hold in large-scale pretraining. However, this warrants further investigation, since we do not fully understand the training dynamics of a LLM.

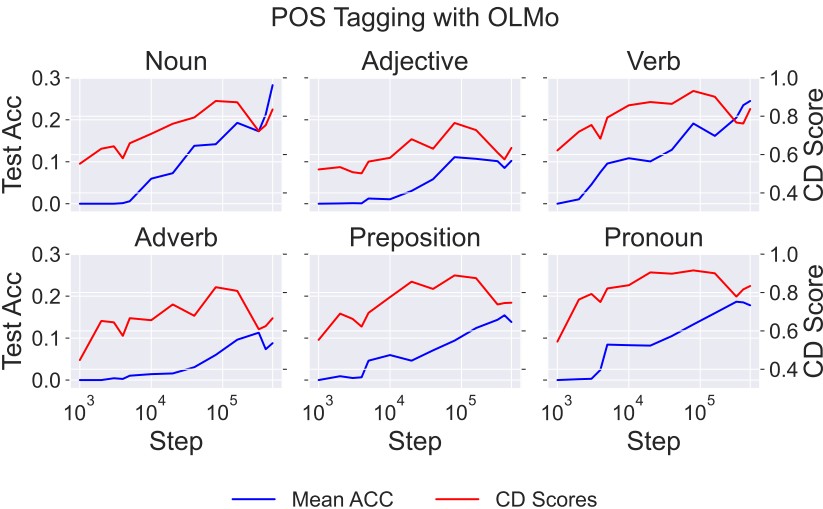

Figure 8: Test accuracy and CD scores of POS Tagging across OLMo-7B (Groeneveld et al., 2024) checkpoints, from 1000 to 500000. .

## B  CONCEPT ENCODING

In this section, we formally define the Concept Encoding and Concept Decoding.

**Definition 1 (Concept Encoding)** *Let $M$ be a transformer model, $\mathcal{Z} = z_1, z_2, \ldots, z_n$ be a set of latent concepts, and $D$ be in-context examples with arbitrary length $K$. A* concept encoding *is an internal mapping $E : \mathcal{D} \to \mathbb{R}^{d_{emb}}$, where $\mathbb{R}^{d_{emb}}$ is the intermediate representation over the model's d-dimensional embedding space.*

**Definition 2 (Concept Decoding)** *Given a transformer model $M$ with concept encoding $E$, a* concept decoding *is a transformer's behavior that there exists a simple function $G$ that can recover the original latent concept and condition the algorithm:*

$$G : \mathbb{R}^{d_{emb}} \to \mathcal{Z}$$

ICL performance of given $z$ is related to how well the decoder $G$ can infer the original latent variable $z$. To quantify this, we introduce the notion of *decodability*. For any given decoder, we define decodability as follows:

**Definition 3 (Decodability)** *For a given decoder $G : \mathbb{R}^{d_{emb}} \to \mathcal{Z}$ and a specific latent variable $z$, we define the decodability measures as follows:*

*1. **Accuracy:***

$$A(G, z) = P(G(E(z, D)) = z)$$

2. **_Distribution Similarity_**:

$$S(G, z) = D_f(P(\hat{z}) \| P(z))$$

Our study suggests that in transformers, the encoder $E$ maps distinct latent variables $z$ to separable representations. The model then applies different algorithms based on the inferred $\hat{z}$. This separability suggests that the transformer is inherently biased toward having a simple decoder $G$. In our study, use the kNN classifier for a decoder, accuracy and for score.

# C  SYNTHETIC ICL EXPERIMENT

## C.1  THEORETICAL ERROR BOUNDS IN SPARSE LINEAR REGRESSIONS

It is known that transformers can achieve Bayes-optimal solutions for linear regression problems by implementing least-squares solutions on the prior of weight sampling (Garg et al., 2022; Raventós et al., 2024). The least-squares estimation of linear regression with a Gaussian prior for task weights can be performed using ridge regression. In the presence of sparsity, the least-squares solution can be obtained through lasso regression with optimal weight searching. The error bounds of our task depend on whether the underlying basis is discovered by the model. We consider two extreme cases:

1. If the model is incapable of inferring any basis in $\mathcal{B}$, it would perform a $D$-dimensional regression with $r$-sparsity, where $D$ is the total dimension and $r$ is the number of non-zero elements.

2. If the model is capable of inferring the basis in $\mathcal{B}$, it can perform an $r$-dimensional regression adjusted for the corresponding non-zero elements of the inferred basis. In this case, the model could benefit from the tighter $r$-dimensional regression bound.

The possibility of diverse algorithms and corresponding error changes enables us to track the Bayesian inference behavior of the model in a more detailed way. In the following results, we indeed observe a transition from $D$-dimensional regression to $r$-dimensional regression, accompanied by changes in the representations of tasks for each basis.

## C.2  EXPERIMENTAL DETAILS

**Mixture of Sparse Linear Regression.**  We adapt the conventional linear regression setup from Garg et al. (2022); von Oswald et al. (2023) to create latent bases $\boldsymbol{B}$ that we can interpret far more easily than $W$. We study this setting with $D = 16$ dimensional with up to $K = 20$ in-context examples. Each $B_i$ has a rank of 4 and is orthogonal with each other. We independently sample $W$ and $x_i$ for each new input sequence from $N(0, \boldsymbol{I_D})$ the noise $\epsilon \sim N(0, 0.01)$. We add the sparsity constraints to the linear regression task to introduce the latent concept of sparsity basis $B$ that is easily interpretable and analyzable in their representations. With the sparsity constraints, we construct the graphical model $B \rightarrow W \rightarrow Y \leftarrow X$. This construction allows us to visualize the representations of each of the bases (latent concepts in this graph) by aggregating the representations across a set of $W$ and $(X, Y)$ pairs.

**Model.**  We use a 12-layer GPT-2 (Radford et al., 2019) architecture transformer, as implemented by HuggingFace (Wolf et al., 2020). This model is parameterized with an embedding dimension of 256 and 8 attention heads and hasa total of 9.5M parameters.

**Training.**  We train the model with a batch size of 128 for 80K training steps. We use the Adam optimizer (Kingma & Ba, 2017) with a learning rate of 1e-4 and betas of 0.9 and 0.9999. We use a MSE loss over the sequence and only compute the losses on the prediction $\hat{y}_i$.

**Evaluation.**  We construct a test dataset of 1K samples and evaluate the model on MSE loss for the predictions $\hat{y}_i$ along the sequence.

**Compute.**  We use an A100 GPU with 80GB of VRAM. To train these models, it takes about $\sim 8$ hours.

### C.3 Additional results

**Replicate experiments** Here, we run the different seeds of synthetic experiments in Figure 1, and we report the results in figure 9. We observe that a single basis produces distinct loss trajectories for Seeds 1 and 2 as in Figure 1, while Seed 3 demonstrates a consistent loss descent across basis.

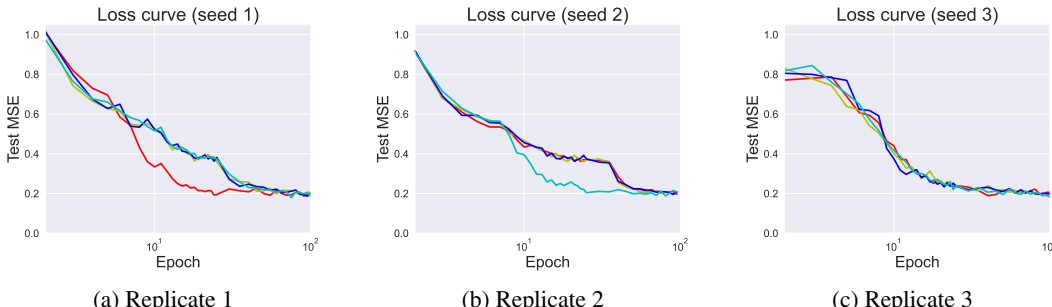

(a) Replicate 1        (b) Replicate 2        (c) Replicate 3

Figure 9: Results from three replicates of experiments corresponding to Figure 1. Each subfigure shows the loss trajectory by basis by different random seeds.

### C.4 Additional Analysis on Section 3.3

**CD Over Training.** We quantified the CD score for the synthetic experiments shown in Figure 1 , with the results presented in Figure 10 and Figure 11. The CD scores for Basis 1 effectively capture the separation of representations observed at (a). An increase in CD scores correlates with a corresponding drop in MSE, as seen in Figure 1, supporting our hypothesis that the CD score can serve as a predictor for the predictability of CD.

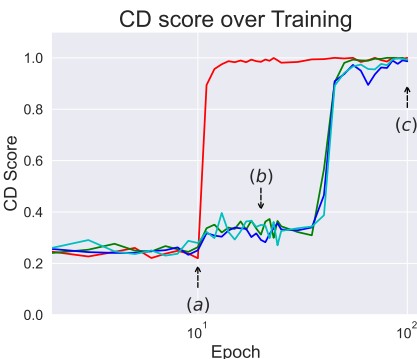

Figure 10: CD score of synthetic experiments in Figure 1 over training. (a), (b), (c) denote the same training points in Figure 1.

**UMAP Over Training.** To analyze how the representations evolve over training across the different layers in the sparse linear regression task, we visualize the UMAP of the representations in Figure 12. We see that concept encoding, the separation of representations by concept, starts to appear at epoch 20 and is only clearly observed from layer 5. Note that the layer index in the figure starts at 0, so layer 4 in the plot equals to what we call layer 5. At convergence, each of the concepts' representations becomes separated from layer 5 and later.

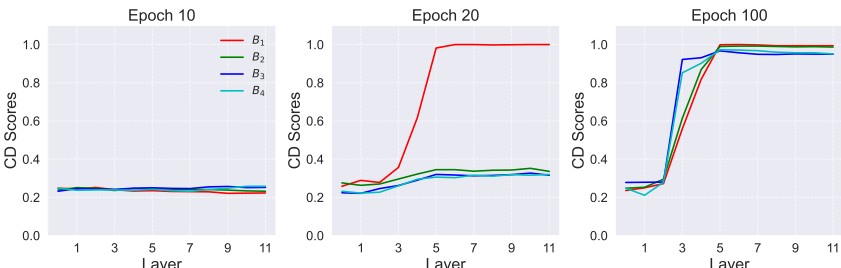

Figure 11: CD score across layers at epoch 10, 20, 100 from the synthetic experiment in Figure 1.

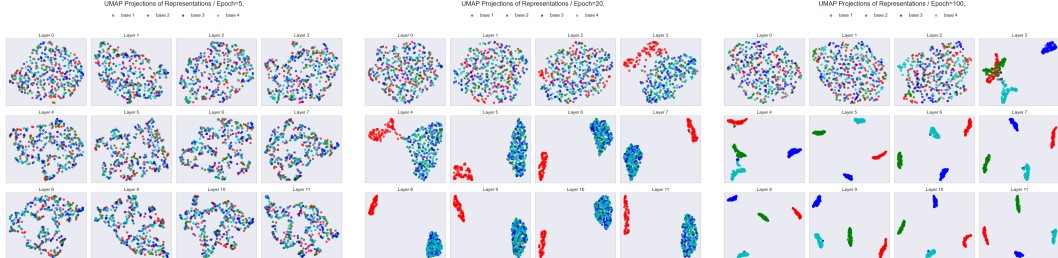

Figure 12: UMAP visualization of representations across the layers over training in the synthetic sparse linear regression task. We visualize the UMAP at epochs 5, 20, and 100 across all the layers. Note that the plot uses zero-based indexing, but we use one-based indexing to refer to the layers in all of the text.

## D   INCREASING COMPLEXITY IN SYNTHETIC EXPERIMENTS

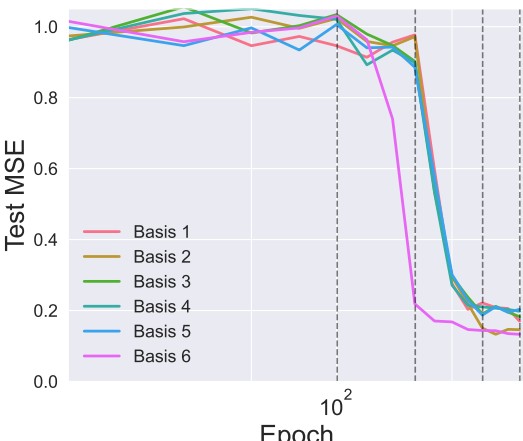

Figure 13: Loss curve over training 300 epochs

### D.1   EXPERIMENT - MORE ORTHOGONAL BASES

We conduct an experiment with 6 orthogonal bases, each spanning 4 dimensions out of 24 total bases. Similar to Figure 1, we observe distinct loss curves over the bases, coupled with clear separation in the representations (see Figure X). Importantly, we observe that basis 6 is learned first (after around 100 epochs), and basis 2 is learned second (after around 200 epochs), while the other four bases are not distinguished by the model until around 300 epochs. Notably, it requires significantly more epochs for the model to learn each concept compared to the scenario in Figure 1 (which uses 4

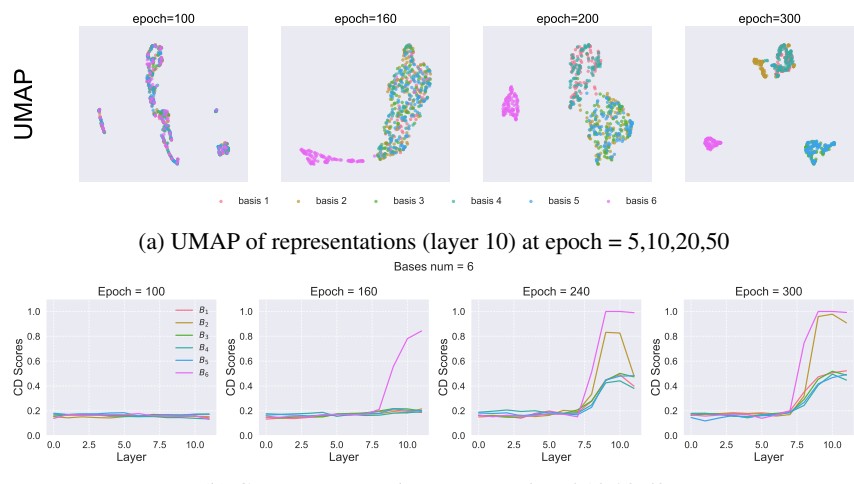

(a) UMAP of representations (layer 10) at epoch = 5,10,20,50

(b) CD score across layers at epoch = 5,10,20,50

Figure 14: Experiment - More orthogonal bases analysis

bases on 16 input dimensions). Following our intuition, it suggests that learning concepts becomes more challenging as the number of concepts increases. Overall, these results support the idea that our proposed concept encoding-decoding mechanism also holds under more complex settings.

## D.2 EXPERIMENT - OVERLAPPING BASES

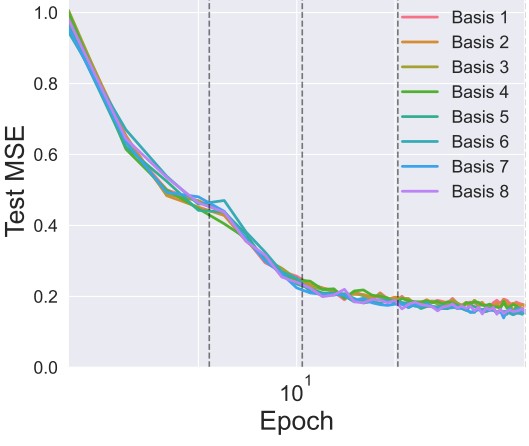

Figure 15: Loss curve over training 50 epochs

We conduct an experiment with 8 overlapping bases, where the first 4 bases (Bases 1, 2, 3, and 4) span 8 dimensions, and the remaining 4 bases span the other 8 dimensions (with a total input dimension of 16). Thus, the first four bases have overlap with another and the second bases have overlap with another. In this setup, we investigate the emergence of separation both within overlapping bases (e.g., within Bases 1, 2, 3, and 4) and between the groups (e.g., between Bases 1, 2, 3, 4 and Bases 5, 6, 7, 8), and examine their relation to subsequent ICL performance.

We observe that the loss curve for each base is identical and undergoes a steep descent around epoch 5 (see Figure D-2 in the link). This loss descent coincides with the separation of the two groups of bases by their representations around epoch 5, while bases within the same group remain entangled and unsorted.

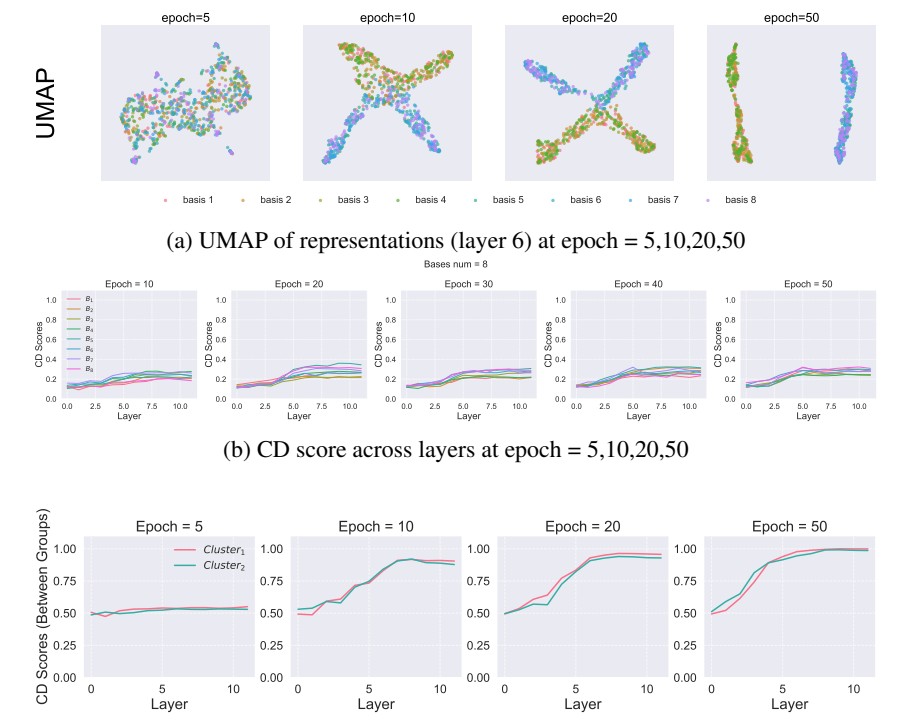

(a) UMAP of representations (layer 6) at epoch = 5,10,20,50

(b) CD score across layers at epoch = 5,10,20,50

(c) CD score between nonoverlapping bases sets(btw basis 1,2,3,4 and 5,6,7,8) at epoch = 5,10,20,50

Figure 16: Experiment - Overlapping bases analysis

These observations suggest several key points. First, the models may not learn to fully separate overlapping concepts, as they can develop shared algorithms to predict the overlapping portions. Second, non-overlapping concepts can be fully separated, which accounts for the significant ICL improvement, as it allows the development of algorithms for orthogonal (non-overlapping) concepts. Third, transformers seemingly learn to classify tasks based on their similarity and associate algorithms at different levels of resolution over the course of training.

# E NATURAL ICL EXPERIMENTS

**Part-of-speech Tagging.** We construct a Part-of-speech (POS) tagging dataset from the English Penn Treebank corpus (Marcus et al., 1994) from the articles of Wall Street Journal. Our POS tags are, Noun, Adjective, Verb, Adverb, Preposition, Pronoun, and Pronoun. We abide by the data-use regulations and, from a total of 4K samples, we filter out sentences that have all 6 POS tags. Then, we split the dataset into a 80-20 train-test split. We evaluate all the models on the test split, and the train split is only reserved for the finetuning experiments.

**Bitwise Arithmetic.** We construct a bitwise arithmetic dataset consisting of 6 different operators: AND, NAND, OR, NOR, XOR, and XNOR. We randomly sample pairs of input binary digits and generate the resulting binary. For training, we construct 10K samples, and, for evaluation, we construct 500 samples.

**Model.** We use a pretrained Llama-3.1-8B model for all of the main natural ICL experiments, if not specified otherwise.

**Training.** For most of the experiments, we do not train the model and only evaluate its ICL performance on the different tasks. However, we only finetune the model in the causal experiments to study the causal relation between the accuracy of concept encoding and ICL task performance. We finetune a model per task family (i.e. POS and bitwise arithmetic). For computationally efficient

finetuning given compute constraints, we use LoRA (Hu et al., 2021), a type of parameter efficient finetuning. We set the rank and alpha to be 16 and the dropout to be 0.1. We train the model on a total of 10K samples with the next-token prediction loss. We only backpropagate the losses on the $\hat{y}_i$ predictions.

**Evaluation.** To evaluate the model's ICL performance, we use greedy decoding to generate answers given different number of in-context examples and compute an exact-match accuracy score – whether the generated sequence is exactly equal to the ground truth.

**Compute.** We use an A100 GPU with 80GB of VRAM for training and inference. Training takes ∼ 4 hours and evaluation takes ∼ 30 minutes for each run.

### E.1 MECHANISTIC INTERVENTION STUDY FROM SECTION 4.1

We present the results for the mechanistic intervention study probing whether different concept encoding triggers different decoding algorithms and whether they are causally related in Figure 17.

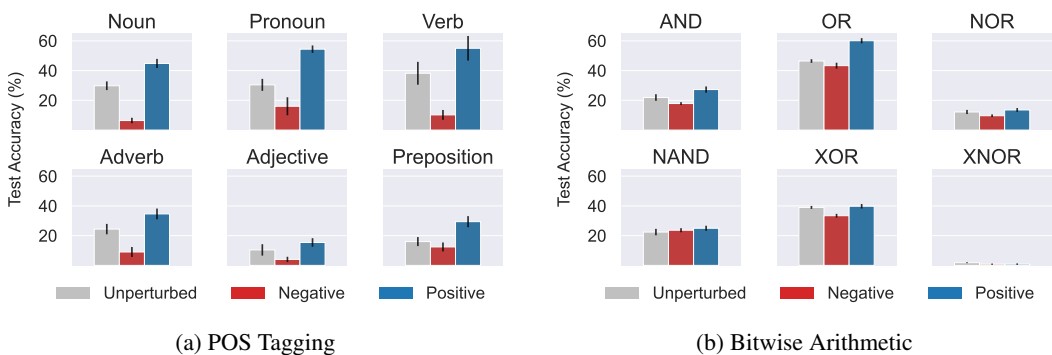

(a) POS Tagging            (b) Bitwise Arithmetic

Figure 17: Causal analysis of concept encoding by intervention. We patch the activations of the input with the correct and incorrect latent concept to demonstrate that the inferred concept embedded in the representation can causally improve or degrade performance. We intervene at layers 15 and 13 respectively for the POS and arithmetic tasks. The results show that the performance is causally dependent on the latent concept representations. Error bars represent the standard deviation across five different replicates of experiments.

### E.2 GENERALIZATION WITH DIFFERENT MODEL FAMILIES AND SCALES

In both the POS and bitwise arithmetic tasks, we observe a positive correlation between CD and ICL test accuracy across different model families and scales. Interestingly, in all of the Gemma-2 family and Llama-3.1 70B models, Noun, Pronoun, and Verb show the clearest signs of concept encoding-decoding behavior, as we saw in the Llama-3.1 8B model in Figure 5. In the bitwise arithmetic task, AND, NAND, OR, and NOR (classes that showed the strongest encoding-decoding behavior in Llama-3.1 8B), also show the strongest signs of concept encoding-decoding behavior across all of these models. Given that many LLMs are trained on similar sources of pretraining data (Soldaini et al., 2024; Gao et al., 2020) (CommonCrawl, Wikipedia, etc), we conjecture that the models may have learned similar encoding-decoding mechanisms for these concepts.

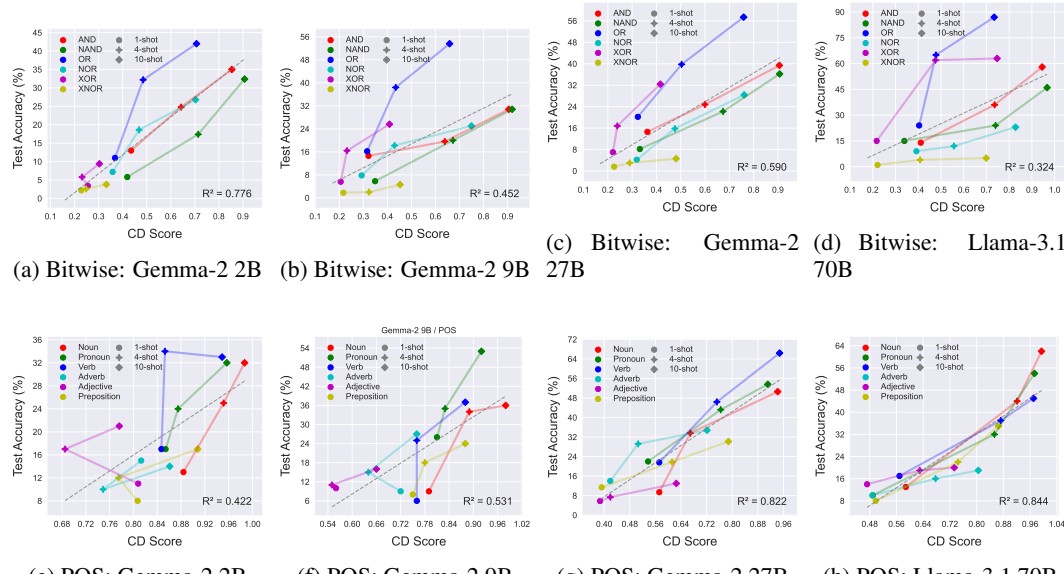

Figure 18: CD score vs ICL performance across Gemma-2 models (2B/9B/27B) and Llama-3.1-70B. The positive correlation between CD and ICL performance seen in Llama-3.1-8B generalizes across different models and scales. The grey dashed lines are linear lines of best fit. These results suggest that the accuracy of concept encoding is closely coupled with downstream ICL performance.

### E.3 PAIRWISE CONCEPT DECODABILITY COMPARISON

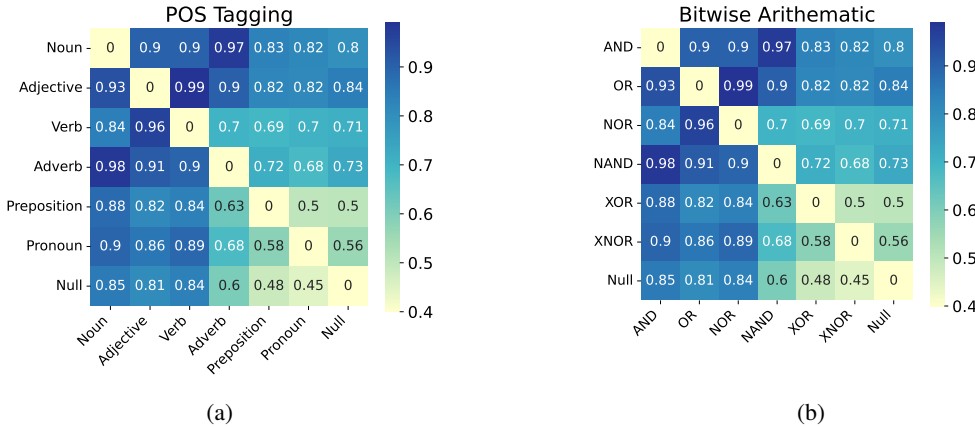

Figure 19: Pairwise CD scores for POS Tagging and arithmetic tasks at 4 shot. Pairwise CD scores identifies the clustered tasks

## F PROMPTING EXPERIMENTS

**Experimental Setup.** To study whether concept encoding is a unifying principle that underlies different mechanisms to improve ICL, we also experiment with prompting. Instead of hiding the concepts and letting the model infer, we include information about the true concept for the examples (e.g., including the true label of AND operator or the instruction of "Find the first noun in the sentence").

**Results.** As discussed in Section 5, we question how prompting may be affecting the concept encoding in increasing task performance. As expected, prompting improves the performance of the model, especially in the bitwise arithmetic experiments. Simultaneously, we observe that the decodability score of the latent concepts also increases drastically. However, we interpret these results with caution because the model may be capturing spurious correlations from the differences in the input distribution. Specifically, the bitwise arithmetic experiments show high decodability even in the beginning layers of the model.

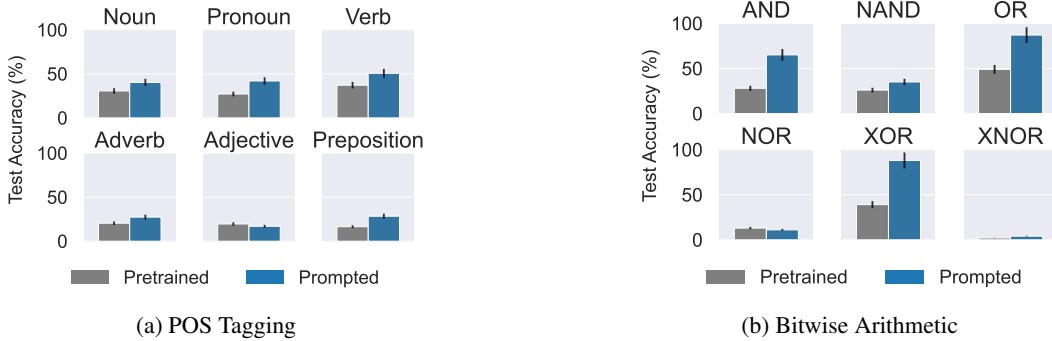

(a) POS Tagging

(b) Bitwise Arithmetic

Figure 20: ICL test accuracy across 12 tasks in POS tagging and bitwise arithmetic with prompts containing the true concept (e.g., AND, "Find the first noun in the sentence") of the task.

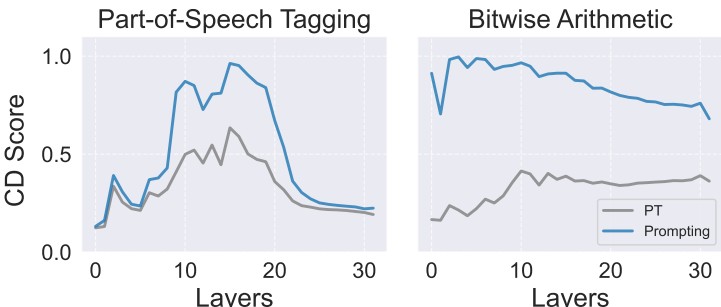

Figure 21: CD score across layers for POS tagging and bitwise arithemetic in Llama-3.1-8B for the prompting experiments. We include the true labels of the latent concept (i.e. "Find the first noun in the sentence."). We detail the experimental setup in Appendix F.