# OpenReview forum: "From Context to Concept: Concept Encoding in In-Context Learning"
_ICLR.cc/2025/Conference — Submitted to ICLR 2025_

### Official Review · Reviewer_HHCg · 2024-10-21

**Soundness:** 3
**Presentation:** 2
**Contribution:** 2
**Rating:** 5
**Confidence:** 4

**Summary:**

The authors show that Transformers learn to solve certain tasks in-context by inferring/embedding contexts in separable representations. Conditioning in these task variables allows the Transformer to accurately predict in-context. The authors show this in experiments conducted both on toy-models trained to perform linear regression and on large-scale Transformers like Llama 8B. They also show that task separability correlates with ICL accuracy. This connection is also demonstrated using activation patching and fine-tuning on selective parts of the Transformer models.

**Strengths:**

* Research question is timely.
* The authors perform multiple experiments both with toy models and LLMs.
* The experiments and analyses are conceptually simple and neatly replicate findings from Hendel et al. 2023.
* Multiple types of interventions were used to validate the efficacy of the task representations, including fine-tuning and activation patching.

**Weaknesses:**

While the experiments and analyses are sound, the results seem more like a replication of the findings from Hendel et al. 2023 or Todd et al. 2023. It is not clear how the concept encoding/decoding framework differs from the Task Vector framework, or why it is necessary to use the concept encoding/decoding framework in the first place. Would it not be fair to characterize the separable representations of the tasks in the ICL experiments as task vectors?

As such, the findings do not seem very novel or surprising in light of previous papers, like Hendel et al. 2023. While the results presented are interesting, I think the paper would benefit a lot from showing something that hadn't already been shown in previous works. For instance, the analysis showing that fine-tuning early layers of Llama improves the concept separability goes in this direction. At the very least, the authors can help explain why the findings are novel, why their framework is needed, or why performing experiments the way they were done improves our understanding of ICL beyond previous papers. Currently, the paper reads like it gives more credible evidence to the existence of task vectors, which is nice, but it choses to call it 'Concept encodings' instead, which is confusing and seems unnecessary.

The writing and explanations can also be improved in various places. The framework that is proposed, which makes reference to 'concepts' is confusing. Why not just stick with existing terminology like task vectors? The term 'Concept' is very loaded, and it is not clear that it adds anything to modelling ICL here.

At the same time I think the paper gives some nice supplementary evidence for the existence of task vectors. I would be willing to increase my score if the authors can address the above criticisms.

There are also some typoes and weird formulations:
* Line 122 "Bayeisan"
* Line 75, 261, 269 "solve ICL" doesn't seem quite right. The models learn to perform ICL, but ICL is not solved.
* Line 161 "over the sequence of sequence of context length 20"
* I don't think the quote at the beginning of the paper adds anything and I would recommend removing it.

**Questions:**

* Why does context decodability peak in the middle layers and go down afterwards?
* In the first experiment, you perform UMAP on the layer activity to find the clusters. UMAP often exaggerates differences. Does kNN classification work here too?
* When you talk about "layer activations", do you mean residual stream representations, or the output of the transformer layers, which are added to the residual stream?

---

### Official Review · Reviewer_ybCA · 2024-10-30

**Soundness:** 3
**Presentation:** 4
**Contribution:** 3
**Rating:** 8
**Confidence:** 4

**Summary:**

This paper studies in-context learning in transformer models through the bayesian lens of concept inference.
They find that in a transformer trained on synthetic data, the model learns to separate tasks in its representation space, and this separation is important for task-specific prediction. They also study how concept encoding and decoding behavior emerges in transformers pre-trained on natural language, and find similar results.

**Strengths:**

The paper is well written and easy to follow, and the figures clearly communicate the experimental results.  The authors test a variety of tasks to show the generality of the findings, though they are relatively simple in nature. There are several lines of evidence that support the paper's conclusions, and it appears there is an appropriate amount of caution in presenting results the authors are unsure about.

**Weaknesses:**

- The tasks studied here are rather simple. When tasks become more complicated, it's unclear whether the task-vector (and thus) concept inference hypothesis will hold. For example, in the synthetic setting, what happens as you increase the number of latent concepts to be learned? Do you find that more latent concepts causes the encoding to become less task-separable?

Some of the experiments lack details that might help clarify some confusion/help with future replication. In particular, I have questions about the experiments in 4.3 and 4.4:

- The experiment described in section 4.3 seems almost identical to the intervention experiment done by Hendel, et al. [1] to validate the "task vector" hypothesis  (at least the positive case), but is missing experimental details. Are there any other differences in this setup besides the tasks, and testing with a "null" task vector? (e.g. do you patch at the final token/multiple tokens,

- For the fine-tuning experiments in section 4.4, how can we be sure that the performance increase is due to the "concept encoding" and not something else? Can you describe your fine-tuning experiment setup in a bit more detail? Is each of these subplots a separate fine-tune, or do you fine-tune the layer set on all tasks at once? There are usually also performance gains for fine-tuning the last 10 layers as well. While not stated, it might be worth clarifying whether this paper's hypothesis for this increase is that fine-tuning the final layers strengthens the concept decoding capabilities of models.

The results could also be strengthened by showing these results hold across other model sizes and model families, since the only pretrained LLMs this paper studies is Llama 3 8B (with some training checkpoints results on OLMo-7B). I'd be curious how separability of the representations change across model sizes of the same family (for example - Llama 8B & 70B), or Pythia (Biderman, et al. [2]). Though, as it stands, the current results are reasonable evidence for the claims made.

___

Minor Notes (not worried about this, but just noticed while reading through):
- Misspelling in Line 297:  "overt" -> "over"
- Mis-capitalization in Line 790: In this section, "We" -> we
- Misspelling in Line 862: "synthetinc" -> synthetic

___
[1] Hendel, et al. In-Context Learning Creates Task Vectors. 2024. (https://aclanthology.org/2023.findings-emnlp.624/)

[2] Biderman, et al. Pythia: A Suite for Analyzing Large Language Models Across Training and Scaling. 2023. (https://proceedings.mlr.press/v202/biderman23a.html)

**Questions:**

- In some recent work,  Mittal, et al. [3] suggest that inferring latent variables doesn't necessarily improve ICL performance, and that the "task vector" view of ICL may be due to parametric shortcuts that are learned by transformers for certain tasks. I'm curious whether this paper's findings complement, support, or contradict this argument.
___
[3] Mittal, et al. Does learning the right latent variables necessarily improve in-context learning? 2024. (https://arxiv.org/pdf/2405.19162)

---

### Official Review · Reviewer_s1LL · 2024-10-31

**Soundness:** 3
**Presentation:** 3
**Contribution:** 2
**Rating:** 6
**Confidence:** 2

**Summary:**

This paper studies the training dynamics of in-context learning (ICL) in transformers by analyzing their representations. It demonstrates the emergence of concept encoding, where the model first encodes the latent concepts in the representation space, and decoding, where the model applies a selective algorithm. They initially show the existence of concept encoding-decoding on experiments with synthetic tasks using a smaller autoregressive model on a mixture of sparse linear regression tasks. The same concept encoding-decoding mechanism exists in the pretrained Llama-3 model, where the authors show that concept encoding holds for POS tagging and bitwise arithmetic tasks as well. The study further shows a causal link between concept decoding capabilities and ICL performance.

**Strengths:**

- **S1:** The paper studies an interesting area of ICL, where the authors propose a new perspective on the training dynamics of ICL by showing the existence of a two-step encoder-decoder mechanism within the transformers.
- **S2:** The paper is well-designed with sound experiments and with the study starting from synthetic and simpler tasks on a smaller model, and extending to the similar trends on a larger, real-world model and NLP tasks. Authors further conduct additional analysis based on model patching and fine tuning.

**Weaknesses:**

- **W1:** While the paper is well written overall, the introduction does not emphasize the main contributions and it is challenging to identify the importance of key insights from the beginning. Next, the paper closes with a brief discussion but lacks a fully rounded conclusion.
- **W2:** Figure 1 is hard to interpret since the same markers are used for different LSE and Lasso regression. Moreover, there is also unnecessary whitespace around Figure 1. Finally, I believe there is a typo in row 377 and the text should refer to Figure 5.
- **W3:** Although the study evaluates both synthetic and real-world tasks, the real-world experiments are limited to a single model (Llama-3.1-8B) and two relatively simple tasks, which raises concerns about whether the concept encoding-decoding mechanism will generalize to more complex or realistic tasks and larger models. Additional experiments on diverse or harder tasks could strengthen the evidence for generalizability.
- **W4:** The paper shows unsurprising and expected results on Figure 3 and Figure 8. The finding that increased number of demonstrations lead to better encoding and decoding seems expected, as more examples provide more “learning” signal, which is observed for few-shot learning problems.
Next, observing that some tasks fail to achieve high accuracy and Concept Decodability (CD) due to representation limitations aligns with existing research about the generalization capabilities of ICL and ICL failing in cases when the new or similar-enough task was not observed so frequently during pretraining, which is commented in the Discussion section.
Finally, the observation that fine tuning the model improves CD and ICL accuracy is not unsurprising as the representation subspaces are aligned and the ICL task is now the same as the IWL task due to fine tuning.

**Questions:**

This work explores an interesting and relevant topic while providing constructive insights into ICL training. However, the authors should improve the paper structurally by having clearer contribution highlights and a more rounded conclusion paragraph. Moreover, the results concerning number of demonstrations, fine tuning and the connection between the CD and ICL accuracy come as unexpected and are hindering the paper's contributions.

Given everything considered, I would be open to raise by score if the authors address the questions stated under the “Weaknesses” section and the following ones:
- In the synthetic experiments in section 3, layer 5 is analyzed closely. Why was this specific layer chosen, and how do observations vary across other layers?
- Can you please provide more details on the training setup for the synthetic experiments in section 3. Do you train 4 different models for different betas, or is it the same model? If it is the same model, how do you perform UMAP?
- Have you considered testing the encoding - decoding capabilities across different models and tasks to show the generality of the encoding mechanism in large language models? Can the same be observed in multi-modal models?
- Have you tested how the model handles more complex tasks, multi step tasks or tasks where concept overlap exists? Could you perform additional experiments to show the scenario with overlapping concepts?

---

> ### Comment · Reviewer_s1LL · 2024-11-25
>
> Thank you for the detailed response and for addressing my concerns.
>
> The paper is now clearer to me, and I appreciate the additional experiments on different models and sizes, as well as the new layer visualizations and overlapping concepts.
>
> After reading other reviews and rebuttals, while acknowledging the concerns raised by others, particularly about similar contributions related to task vectors and fine tuning, I believe the authors have have effectively addressed my concerns so I have decided to raise my score.

---

### Official Review · Reviewer_3fqh · 2024-11-08

**Soundness:** 3
**Presentation:** 2
**Contribution:** 2
**Rating:** 5
**Confidence:** 4

**Summary:**

The authors investigate how concept understanding develops within transformers during training by studying a small model on synthetic tasks designed for in-context learning (ICL). They found that as the model learns to represent different underlying concepts (like identifying parts of a sentence), it also builds ways to decode these concepts, leading to better ICL performance. Examining a larger, pretrained model (Llama-3.1-8B), they show that its ability to encode concepts directly impacts its ICL effectiveness. Using techniques like controlled fine-tuning and targeted interventions, they demonstrate that improving concept encoding helps the model perform better on ICL tasks. They also experiment with prompting, finding that it can enhance concept encoding and ICL performance.

**Strengths:**

- The study addresses an interesting and practical research question: understanding the mechanism behind in-context learning (ICL) in LLMs. This is an intriguing problem from a scientific perspective and has important implications for real-world applications.
- The authors designed a simple yet reasonable synthetic task to explore the model's emergent behavior in concept encoding and decoding. Although straightforward, the task is well-suited to investigate the research question.
- By training a small GPT model from scratch and prompting the Llama 8B model, the authors effectively examined the hidden representations of LLMs, revealing the coupled emergence of concept encoding and concept decoding.

**Weaknesses:**

- The research question and task design draw on prior related work, which may limit the novelty of this work on these points. (But I still want to emphasize that the authors add an interesting twist by incorporating sparsity constraints into the sparse linear regression task, which is a valuable contribution of this work. )
- The hypotheses for each experiment and their specific contributions are not entirely clear. It is difficult to discern what each experiment aims to verify and whether the findings are novel.
For example, in line 146, the authors state, "we demonstrate the emergence of concept encoding and decoding are mutually reinforcing." However, the experimental results lack sufficient evidence to support this claim, which may make this assertion seem overstated. I would encourage the authors to clarify their findings throughout the paper, clearly distinguishing between reproductions of prior work and novel insights, whether in synthetic tasks or large-scale experiments.

**Questions:**

1. What specific hypotheses are being tested in each experiment? Could you clarify these?
2. How does each experiment contribute to the overall research question? Can you make these connections more explicit?
3. Are the findings entirely new, or do they replicate previous results? It would help if you clearly identified which results are reproductions and which are novel insights.
4. In line 146, you mention that concept encoding and decoding are "mutually reinforcing." Could you provide more evidence or context to support this claim? It may currently come across as overgeneralized.
5. How does adding sparsity constraints to the sparse linear regression task enhance the study? Could you explain this addition’s significance in more detail?

---

### Meta-Review · Area_Chair_BsRU · 2024-12-19

**Metareview:**

The authors present an analysis to understand the mechanism of ICL. They propose a concept encoding-decoding mechanism, validated across a few models. The experiments seem well supported, but overall seem relatively incremental given existing work. Expanding this analysis to more in-depth analyses (e.g. how encoding and decoding interact / their relative speed of acquisition / their generalization) and / or to other task families would help make this stronger.

I would strongly recommend a workshop for the paper as is, and to resubmit with a few additional analyses -- the paper is borderline and just falling short of a full conference paper.

**Additional Comments On Reviewer Discussion:**

The authors added experiments to address concerns about generalizability (to models and tasks), and clarity of exposition.

The main remaining issues raised were about overall novelty given past work, whether the work is impactful enough -- about which all reviewers did not agree.

---

### Decision · Program_Chairs · 2025-01-22

Reject